# Construction and Characterization of a Botrytis Virus F Infectious Clone

**DOI:** 10.3390/jof8050459

**Published:** 2022-04-29

**Authors:** Laura Córdoba, Ana Ruiz-Padilla, Julio Rodríguez-Romero, María A. Ayllón

**Affiliations:** 1Centro de Biotecnología y Genómica de Plantas, Universidad Politécnica de Madrid (UPM)-Instituto Nacional de Investigación y Tecnología Agraria y Alimentaria (INIA), Campus de Montegancedo, Pozuelo de Alarcón, 28223 Madrid, Spain; laura.cordobagarcia@gmail.com (L.C.); ana.ruizp@upm.es (A.R.-P.); julio.rodriguez.romero@upm.es (J.R.-R.); 2Departamento de Biotecnología-Biología Vegetal, Escuela Técnica Superior de Ingeniería Agronómica, Alimentaria y de Biosistemas, Universidad Politécnica de Madrid (UPM), 28040 Madrid, Spain

**Keywords:** *Botrytis cinerea*, infectious clones, mycovirus, Botrytis virus F, biological control, functional fungal genomics

## Abstract

Botrytis virus F (BVF) is a positive-sense, single-stranded RNA (+ssRNA) virus within the *Gammaflexiviridae* family of the plant-pathogenic fungus *Botrytis cinerea*. In this study, the complete sequence of a BVF strain isolated from *B. cinerea* collected from grapevine fields in Spain was analyzed. This virus, in this work BVF-V448, has a genome of 6827 nt in length, excluding the poly(A) tail, with two open reading frames encoding an RNA dependent RNA polymerase (RdRP) and a coat protein (CP). The 5′- and 3′-terminal regions of the genome were determined by rapid amplification of cDNA ends (RACE). Furthermore, a yet undetected subgenomic RNA species in BVF-V448 was identified, indicating that the CP is expressed via 3′ coterminal subgenomic RNAs (sgRNAs). We also report the successful construction of the first BVF full-length cDNA clone and synthesized in vitro RNA transcripts using the T7 polymerase, which could efficiently transfect two different strains of *B. cinerea*, B05.10 and Pi258.9. The levels of growth in culture and virulence on plants of BVF-V448 transfected strains were comparable to BVF-free strains. The infectious clones generated in this work provide a useful tool for the future development of an efficient BVF foreign gene expression vector and a virus-induced gene silencing (VIGS) vector as a biological agent for the control of *B. cinerea*.

## 1. Introduction

Mycoviruses infect a large number of diverse fungal species, including most of the plant-pathogenic fungi [1,2,3,4,5]. While sharing some characteristics with animal and plant viruses, mycoviruses have unique characteristics. Most of them lack an extracellular route for infection, are intercellularly transmitted through hyphal anastomosis, and lack movement proteins [5,6]. Recently developed metagenomic approaches have contributed to the identification of an increasing number of novel mycoviruses, which has greatly increased our understanding of their evolution, ecology, and the interactions with their fungal hosts. Mycoviruses reported to date mostly contained positive-sense single-stranded RNA ((+)ssRNA) and double-stranded RNA (dsRNA) genomes, although negative-sense, single-stranded RNA ((–)ssRNA), and single-stranded DNA (ssDNA) mycoviruses have also been described [7,8,9,10,11,12,13].

Most mycoviruses are usually associated with latent infections in their fungal hosts, however, some of them cause dramatic changes including attenuation of fungal pathogenesis (hypovirulence). This trait makes them good candidates for the development of biocontrol strategies against fungal plant diseases. As an example, Cryphonectria hypovirus 1 (CHV1) has been successfully used for the biocontrol of chestnut blight fungus *Criphonectia parasitica*, significantly reducing the impact of this disease in Europe [14,15,16].

Although mycoviruses were described in fungi 60 years ago, most studies of their phenotypic effects have been assessed by natural transmission between fungal isolates by hyphal anastomosis, while reverse genetics systems and in vitro transfection methods have been developed only for a few of them. The use of the infectious clones has been key to get a better understanding of the viral infection mechanisms and to get insight into the fungal–virus interaction at the molecular level, along with the possibility of using mycoviruses as vectors to introduce genes deleterious to the fungal host. Infectious clones constructed for mycovirus Cryphonectria hypovirus 1 [17], Sclerotinia sclerotiorum hypovirus 2 [18] and Diaporthe RNA virus 1 [19] have been essential to demonstrate their cause-and-effect relationship for the hypovirulence in *Criphonectia parasítica*, *Sclerotinia sclerotiorum*, and *Diaporthe perjuncta*, respectively. Furthermore, reverse genetics developed for the CHV1 has provided the ability to identify viral determinants responsible for modulation of host functions [20]. Infectious clones for Saccharomyces 23S RNA narnavirus [21], and Saccharomyces 20S RNA narnavirus [22] have been as well successfully constructed and have provided important information to understanding the molecular basis of narnavirus replication and its persistence in yeast. Reverse genetic studies have also been determinant in investigating the evolutionary relationship between two RNA viruses, yado-kari virus 1 and yado-nushi virus 1 in a phytopathogenic fungus *Rosellinia necatrix* [23]. Most recently, Wang et al. have reported the construction of a full-length clone of the novel ourmia-like virus 4 (SsOLV4), isolated from *Sclerotinia sclerotiorum*, and have demonstrated that only the RdRP-coding RNA segment is sufficient for SsOLV4 replication initiation and transmission [24].

One of the fungal pathogens that has the highest impact on crop production worldwide is *Botrytis* spp. In particular, *Botrytis cinerea* is one of the most widespread and destructive necrotrophic plant pathogenic fungus causing gray mold disease in more than 1400 plant species worldwide, including important crops, among which grapevine is one of the most affected [25]. Although post-harvest appears to be the most critical stage for the development of the disease, the fungus is also a serious pathogen of growing plants, leading to high economic losses. The application of fungicides is currently the common method to control *B. cinerea*. However, fungicide resistance is becoming a problematic issue [26,27] which has highlighted the need for improved crop protection measures and the development of new environmentally friendly alternative methods. Similar to other plant pathogenic fungi, mycoviruses are prevalent in the Botrytis population. RNA_Seq analysis has revealed a complex virome in the Botrytis population, including taxonomically unique virus genera. Previous reports by our group [10,28,29,30,31] have documented the presence of replicating mycoviruses in *B. cinerea* fungal isolates with different types of genomes: dsRNA such as Botrytis cinerea mycovirus 4 [10,32]; (+)ssRNA), belonging to the recently accepted family *Botourmiaviridae* [33], Botrytis ourmia-like virus, and Botrytis virus F (BVF) [28,29,34]; (–)ssRNA, as for example, Botrytis cinerea negative-stranded RNA virus 1 [30], Botrytis cinerea bocivirus 1 [10] and Botrytis cinerea mymonavirus 1 [35]. Many *Botrytis* spp. mycovirus have been determined to induce fungal virulence attenuation. In this group are included, for example, Botrytis cinerea mitovirus 1 [36,37], Botrytis porri botybirnavirus 1 [38], Botrytis cinerea RNA virus 1 [39], Botrytis cinerea CCg378 virus 1 [40], Botrytis cinerea hypovirus 1 and Botrytis cinerea fusarivirus 1 [41].

Among *Botrytis* spp. infecting mycoviruses, BVF has been identified in isolates from different regions around the world, including Spain [10,29]. BVF is a flexuous rod-shaped mycovirus that belongs to the genus *Mycoflexivirus*, in the family *Gammaflexiviridae*. It contains a (+)ssRNA genome of 6827 nucleotides with a 5′ end putative capped and a poly(A) tract at the 3′ end. Sequencing analysis revealed the presence of two potential open reading frames (ORFs), ORF1 encodes for a replicase, with the three conserved replicase regions containing the methyltransferase, helicase and RdRP domains, and the last one is directly translated by an opal putative readthrough codon between the helicase and RdRP regions [34]. ORF2 encodes the viral CP which has a high homology with the most conserved central core regions of the CPs of potex-like plant viruses [34]. The CP is presumably transcribed as monocistronic subgenomic mRNA (sgRNA), although this strategy has not been experimentally confirmed yet.

Despite infectious clones for plant viruses closely related to BVF, included in the *Flexiviridae* family have been reported [42,43,44,45,46], reverse genetic systems based on BVF genome have not been developed yet. Since BVF is a small (+)ssRNA virus and has no effects on the fungal growth rate or virulence of *B. cinerea*, the development of an infectious clone of BVF would be highly feasible and useful in identifying determinants responsible for pathogenesis and those involved in fungus-plant interaction. To build up an effective viral vector based on the BVF genome, it will be a key to also gain insight on the strategies used by the virus for the expression and regulation of its downstream genes.

In the present study, we have analyzed the complete sequence of a BVF strain isolated from *B. cinerea* collected from grapevine fields of Spain and identify a subgenomic RNA species in BVF. We also report the successful construction and molecular characterization of an infectious full-length clone of BVF and analyzed potential alterations of fungal colony morphology, growth rates and virulence on plants. The full-length clone derived from this research work provide a powerful tool for better understanding the mechanisms of BVF infection.

## 2. Materials and Methods

### 2.1. Fungal Strains and Culture Conditions

*B. cinerea* V448, a strain obtained from grapevine fields of Castilla-León (Spain) infected with BVF [29], was maintained on potato dextrose agar (PDA) plates at 23 °C in darkness. For total RNA extraction, fungus was grown in potato dextrose broth (PDB) at 23 °C in darkness. *B. cinerea* B05.10 (model strain isolated from grapevine) and Pi258.9 (isolated from pepper in South Spain) [28] strains were cultured in the same conditions as *B. cinerea* V448 strain and used to test the infectivity of the BVF clones. B05.10 is a fungal strain free of mycovirus, whereas Pi258.9 is naturally infected by two mycoviruses, an ourmiavirus and an umbra-like virus. For the obtention of fungal single spore (monosporic culture), a conidia suspension from sporulated cultured fungus was prepared and diluted to a concentration of 1 × 10^3^ spores/mL. 0.1 mL of this suspension was placed on Petri dishes with water agar (2%) medium and incubated at 24 °C for 16 h. After incubation, germinated spores were individually selected, transferred onto individual Petri dishes with PDA medium, and incubated at 24 °C during six days.

### 2.2. RNA Next Generation Sequencing (NGS) and In Vivo Detection of Mycoviruses

A mix of total RNA isolated from three monosporic isolates of the *B. cinerea* strain Pi258.9 was sent to Macrogen company (Seoul, Republic of Korea) for library preparation (Illumina TrueSeq) and sequence analysis with Illumina NovaSeq 6000, for this sample more than 100 million pair ended reads, 150 bases long were retrieved. Bioinformatic analysis was performed as described by Ruiz-Padilla et al. [10]. Mycoviruses were detected in the infected strain by RT-PCR with specific primers included in Table 1. The transcriptomic data for B05.10 was downloaded from the NCBI SRA database (accession number: SRR7894626). To detect the virus in this reference strain of *B.*
*cinerea,* a bioinformatics analysis was performed as before [10]. Any RNA was detected as a virus using the nr NCBI database.

### 2.3. Determination of 5′ and 3′ Ends, Full Length Genome of Botrytis Virus F Genome, and the Transcription Initiation Site of Capsid Protein (CP) Subgenomic RNA (sgRNA)

BVF genome ends were determined by Rapid Amplification of cDNA ends, RLM-RACE kit (Thermofisher, Waltham, MA, USA). For 5′ termini determination, total RNA from *B. cinerea* strain V448, naturally infected with BVF, was used. The 5′ end sequence was obtained by a nested PCR using two specific primers, BVF 5out and BVF 5in (Table 1), in combination with 5′ outer and 5′ inner RACE primers provided by the kit. Since the BVF viral genome is polyadenylated at the 3′, determination of the 3′ terminus cDNA was directly generated by a reverse transcription using the 3′ RACE adapter provided by the kit. The sequence corresponding to the 3′ terminus was amplified in a nested PCR using specific primers, BVF 3out and BVF 3in (Table 1), in combination with 3′ outer and 3′ inner RACE primers. PCR products were excised from agarose gel, purified, cloned into pJet2.1 vector using the CloneJet PCR Cloning kit (Thermofisher) and sequenced with pJET1.2 forward and pJET1.2 reverse sequencing primers included in the kit. Once the 5′and 3′terminal sequences were determined, full-length BVF genome was sequence by primer walking. Primers used for this purpose are listed in Table 1.

BVF RNA was used to map the transcription start site of the sgRNA corresponding to the CP coding sequence using the 5′ RACE system (Thermofisher). The gene specific primer BVFsg_5OUT was used to synthesize the first-strand cDNA (Table 1). The first-strand was amplified using Taq DNA polymerase (Sigma-Aldrich) and an abridged anchor primer (supplied with kit) together with gene specific nested primer BVFsg_5IN (Table 1). The 5′ RACE products were ligated into pJet1.2 vector (Thermo Scientific, Waltham, MA, USA) and the inserts were sequenced with pJET1.2 forward and pJET1.2 reverse sequencing primers.

### 2.4. In Silico Prediction of RNA Secondary Structure

Computer models of RNA folding were generated using the most recent version of a software package for RNA secondary structure prediction and analysis MFOLD [47]. Thermodynamic parameters applied were those set by Turner et al. [48,49,50].

### 2.5. Construction of a Full-Length cDNA Clone of BVF

cDNA of the full-length BVF genome was synthesized from total RNA by using Superscript IV reverse transcriptase (Invitrogen) with the primer BVF_NotIEcoRI Rev (Table 1). The entire viral genome (6.8 kb) was amplified with the CloneAmp HIFI PCR mix (Takara) using the primers BVF_XbaI Fw and BVF_NotIEcoRI Rev (Table 1). The forward primer BVF_XbaIT7 Fw contained an engineered XbaI site and the T7 promoter sequence followed by part of the BVF 5′ end sequence of the viral genome. The reverse primer BVF_NotIEcoRI Rev introduced a poly (A)_18_ tail and NotI and EcoRI sites at the 3′ end of the viral genome for linearization and cloning purposes. The resultant PCR product was gel purified with NucleoSpin Gel and PCR Clean-up (Takara, San Jose, CA, USA) and ligated into pUC19 vector between XbaI and EcoRI sites by using the T4 DNA ligase (ThermoScientific, Waltham, MA, USA) to generate the clone pUC19-BVF. Recombinant plasmid DNA were extracted and purified using the Zyppy plasmid miniprep kit (Zymo Research, Irvine, CA, USA) and checked for the presence of the BVF genome by restriction digestion and sanger sequencing. The strategy followed to construct the full-length cDNA clone of BVF is shown in Figure 3A.

### 2.6. Transfection of Botrytis cinerea Protoplasts with BVF RNA Transcripts

The plasmid DNAs from positive BVF clones were linearized with NotI. The capped RNA transcripts were synthesized using a MegaScript T7 Kit (Invitrogen, Massachusetts, USA) following the manufacturer’s recommendations, and an optimized 4/3 ratio of methylated cap analogue to GTP for direct inoculation into *B. cinerea* protoplasts. Two different strains of *B. cinerea*, B05.10 and Pi258.9, were used. A 3 to 4 day-old 200 mL PDB culture of mycelia was used to generate fungal protoplasts following the methodology of Antal Z. et al. [51] with some modifications. Briefly, 1 g of mycelia were collected by filtration of the media through miracloth and incubated with 20 mL of washing buffer (0.6 M KCl, 0.1 M phosphate buffer pH 5.8) and 200 mg of VinoTaste Pro enzyme (Vinotaste Pro, Lamothe Abiet, Canéjan, France). The suspension was maintained for 2.5 h in a water bath at 30 °C and 50 rpm. Resultant protoplasts were collected by filtration (nylon 30–40 mm), washed, centrifuged at 1200× *g* for 15 min, and suspended in cold KC buffer (KCl 0.6 M, CaCl_2_-2H_2_O 50 mM, pH 6) to a concentration of 2 × 10^8^ spheroplast/mL. Ten microliters of transcription reaction were gently mixed with 100 μL of spheroplasts in 85 µL of buffer KC and incubated on ice for 30 min. PEG4000 (25% PEG4000, 50 mM CaCl_2_-2H_2_O in 10 mM Tris buffer, pH 7.4) was added with gentle mixing and samples were then left for 25 min at room temperature. Following addition of 1.2 mL cold KC buffer, gentle mix and centrifugation (5 min, 1200× *g*), the protoplasts were suspended in 2 mL cold KC buffer and placed on the center of a 9 cm petri dish and mixed with 20 mL of SH medium (NaNO_3_ 1 mM, Sucrose 0.6 M in 5 mM TRIS buffer pH 6.5, agar Oxoıd 1.5%). After 6 days of incubation at 23 °C total mycelium from each plate was collected using 2 mL of sterile water and equally distributed into PDA and PDB plates. After three passages PDB inoculated cultures were checked for BVF infection.

### 2.7. Total RNA Extraction and RT-PCR for BVF Detection

The fungal mycelium was harvested from liquid culture and dried using Miracloth and sterile filter paper and frozen at −80 °C. Total RNA was extracted from 1 g of mycelia using the NZYol (NZYTech, Lisbon, Portugal) following manufacturer’s recommendations. Total RNA was quantified and run on an agarose gel for quality determination. Transfected fungal cultures grown in PDB medium for 5–6 days at 23 °C were analyzed by RT-PCR for BVF detection. RT-PCR was performed with the NZY First-Strand cDNA Synthesis Kit (NZYTech) and the Supreme NZYTaq *II DNA* polymerase (NZYTech) using primers BVF_FwQ1 and BVF_RevQ2, and BVF 5′ end XbaI Fw and NewBVF 738 Rev (Table 1). The RT-PCR products were examined by ethidium bromide agarose (1% or 2%) gel electrophoresis.

### 2.8. Northern Blot Hybridization of BVF RNAs

RNA hybridizations were performed following procedures recommended by Sambrook and Russell [52]. Twenty micrograms of each RNA sample were separated by electrophoresis an agarose–formaldehyde gel (12 g/L agarose), transferred by capillarity onto a Nylon (Hybond-N) membrane, and probed with the cloned BVF segment for CP (868 bp) or with RdRP segment (557 bp) labeled with P_32_-dCTP. The labelling reaction was performed using random oligonucleotide primers and cloned exonuclease-free *E. coli* DNA polymerase I, Klenow fragment. Northern blot hybridization was performed overnight to 65 °C in phosphate buffer.

### 2.9. Determination of Mycelial Growth and Virulence Assays

Mycelial agar plugs (6-mm-diameter) were removed from actively-growing colony margins of 3-day-old PDA cultures of BVF-infected and BVF-free *B. cinerea* strains and transferred to the center of PDA petri dishes (5 replicates) and incubated in the dark at 23 °C. Colony diameters were measured after 2 to 15 days to determine the colony morphology, the radial growth rate (RGR), and number of spores. RGR (cm/day) = (D3–D2)/2, where D2 and D3 represent the diameter of 2- and 3-day-old colony, respectively, was calculated [37]. To evaluate the virulence, agar plugs from actively-growing colony margins of 3-day-old PDA cultures of BVF-infected and BVF-free *B. cinerea* were placed on detached leaves of tomato (*Solanum lycopersicum)* and pepper (*Capsicum* *annuum*) plants with the mycelial side facing the leaf surface. Four leaves were inoculated with two plugs each and placed on the bench at room temperature for 72 h. Two perpendicular measurements of lesion diameter on each inoculated leaf were averaged. Each assay was performed two or three times. Data were analyzed using a one-way analysis of variance (ANOVA).

## 3. Results

### 3.1. Determination of the BVF Genome Ends and Transcription Initiation Site of CP sgRNA

Although the genomic coding regions of several BVF strains have been completely sequenced [29,34,53], the sequence of the 5′UTR and 3′UTR has been reported only for two BVF strains [34,53]. The genome sequence of BVF-V448 was previously determined by NGS (Accesion number LN827953) [29]. In order to determine the complete sequence of the mycovirus, the RLM-RACE technique was used to obtain the 5′ and 3′ terminal sequences. Two single products of the expected size (about 200 bp) corresponding to the 5′ and 3′ terminal sequence were obtained by nested RT-PCR using the primers shown in Table 1.

The 5′ terminus of BVF-V448 contained 63 nucleotides preceding the predicted start codon of ORF1, and it was 13 nt longer at the 5′ end compared to the genomic sequence obtained by NGS (LN827953) [29] and showed an identity of sequence of 96%. The 3′ terminus had 71 nucleotides excluding the poly A tail, and the sequence was 100% identical to LN827953. Both termini were also aligned with the corresponding sequences of the other two BVF strains available in the Genbank nucleotide sequence database (Accession Numbers: AF238884 and MH338171) [34,53]. Sequence MH338171 was found to contain one additional G at the extreme 5′ terminus compared to the other BVF strains (Figure 1A). The 5′ terminal sequence data of BVF-V448 showed variations with respect to the BVF strains AF238884 and MH338171, with identity values of 87% and 92%, respectively. However, the 3 ‘terminal sequences of the different strains of BVF showed higher identity values (Figure 1A), BVF-V448 3′ terminal sequence showed 100% identity with AF238884 and 97% identity with MH338171. Thermodynamic modeling of the BVF minus-strand sequence predicted that the 40 nucleotides of the 5′ terminus fold into a putative simple stem-loop (SL) structure, with a free energy value of −1.20 Kcal/mol (Figure 1B). BVF positive-strand 3′ terminal sequence (last 40 nt) was also found to fold into a putative SL structure with a free energy value of −5.70 Kcal/mol (Figure 1B).

Complete sequence of full-length BVF-V448 cloned in this work was determined by primer walking (GenBank Accession number OM928008). The sequence of BVF-V448 was 6827 nt in length and showed identity values of 92%, 87% and 84% to BVF strains published by Donaire and Ayllón (LN827953), Howitt et al. (AF238884) and Svanella–Dumas et al. (MH338171), respectively. At the amino acid level, BVF-V448 RdRP shared 97, 95, and 91% identities with strains LN827953, AF238884 and MH338171, respectively, while amino acid sequences of BVF-V448 CP shared 99% with strain LN827953 and 96% identity with strains AF238884 and MH338171.

To determine the transcription start site of the BVF CP-sgRNA, 5′ RLM-RACE specific for capped RNA was performed. Sequencing analysis revealed that the 5′ end sequence of all clones started with a guanine (G) at nt 5822 followed by an adenylate region (Figure 2A), suggesting that the transcription start site of the CP-sgRNA is located within the intergenic region, starting 67 nts downstream the end of the RdRP ORF and 26 nts upstream of the CP ORF. Comparison of the BVF CP sgRNA sequence with those previously published (LN827953, AF238884 and MH338171) showed that all of them contained a conserved sequence around the transcription start site that could play a critical role in the regulation of transcription of CP-sgRNAs (Figure 2A).

BVF belongs to the genus *Mycoflexivirus*, family *Gammaflexiviridae*, order *Tymovirales*, and similar sequences were also found in putative promoter sequences for the sgRNAs among other viruses in the families of the order *Tymovirales*, as *Alphaflexiviridae*, *Betaflexiviridae*, *Tymoviridae*, including viruses in the genus *Capillovirus*, *Carlavirus*, *Trichovirus*, *Citrivirus*, *Vitivirus*, *Potexvirus* (Figure 2B), with a highly conserved hexanucleotide around the transcription start sites of sgRNAs, which suggests that this sequence may be key for the regulation of sgRNA synthesis in all these viruses. The secondary structure around the transcription initiation site of CP-sgRNA was predicted by analyzing a 30 nt region, which extended from positions −11 to +19 from the transcription start site in the minus sense strand. This sequence was predicted to fold into a SL structure with the transcription initiation site located near to the loop region (Figure 2C).

### 3.2. Infectivity of Full-Length BVF Clone in Botrytis cinerea

A full-length cDNA clone of the Spanish BVF strain was constructed (Figure 3A). Synthetic capped RNA transcripts were generated from three full-length BVF cDNA clones linearized with NotI by using T7 RNA polymerase. Capped RNA transcripts were mechanically inoculated into protoplasts of *B. cinerea* B05.10 and Pi258.9 strains. Transfection in B05.10 and Pi258.9 was performed, and after three consecutive passages, transfected mycelia was collected and subjected to RNA extraction and RT-PCR for BVF detection. BVF viral RNA was detected (amplicons of approximately 300 bp and 800 bp in CP and RdRP regions, respectively) in both, B05.10 and Pi258.9 transfected mycelia, whereas viral RNA was not detected in mock fungi as expected (Figure 3B). Monosporic subcultures of transfected B05.10 and Pi258.9 cultures were established to assess the efficiency of BVF vertical transmission. Among the 12 monosporic isolates obtained, BVF RNA was detected in 91.6% of the B05.10 isolates and in 100% of Pi258.9 isolates (Figure 3B and data not shown). These results indicate that full-length BVF clone is infectious through repeated subculturing, able to stablish a robust and stable infection in both fungal strains, B05.10 and Pi258.9, and be efficiently vertical transmitted.

The analysis of the public transcriptomic data from B05.10 [54] indicated that this isolate was not infected with viruses, then it can be used as a mycovirus recipient in assays to determine the influence of viral infection on the phenotype and virulence of the fungus. However, our NGS results showed that the three monosporic isolates of *B. cinerea* Pi258.9 strain could be infected with three possible variants of the mycoviruses Botrytis cinerea ourmia-like virus 2 (BcOLV2) isolate BCS12_DN83 (MN605468 [10], 94% identity at nt level), Sclerotinia sclerotiorum ourmia-like virus 4 isolate SsOV4/SX276 (MT706021 [24], 97% identity at nt level) and Sclerotinia sclerotiorum umbra-like virus 3 isolate (SsULV3) [55] (MF444274, 94% identity at nt level). The RT-PCR results indicated that the monosporic isolate selected to be used as the recipient of the infectious clone BVF was infected with the BcOLV2 isolate BCS12_DN83 (MN605468) and SsULV3 (MF444274) (data not shown). 

To confirm the replication of the infectious clone BVF-V448 and analyze the production of a CP-sgRNA, total RNA preparations from *B.*
*cinerea* B05.10 and Pi258.9 strains transfected by the infectious clone BVF-V448 and monosporic strains were analyzed by Northern blot hybridization. Hybridization with the specific BVF CP probe allowed visualization of the BVF genomic RNA and a putative sgRNA of approximately 1.1 kb in the samples infected with BVF-V448 clone (Figure 4). Additionally, other bands of different sizes were visualized between genomic and subgenomic RNAs and under sgRNA, probably corresponding to defective RNAs (D-RNAs). The presence of D-RNA has been previously described by Howitt and coworkers [34]. The presence of genomic RNA and D-RNA was confirmed by hybridization with a specific BVF RdRP probe, however, no bands of the CP-sgRNA size were observed (Figure 4). The results obtained confirm the replication of the BVF-V448 infectious clone in both *B.*
*cinerea* strains, indicated that the mycovirus synthesizes a sgRNA to translate the CP gene, and suggest that D-RNA generation is common during the replication of the mycovirus.

### 3.3. Effect of BVF on Fungal Growth and Virulence

To assess the biological effects of BVF-488 infection, both in vitro and in vivo, the growth rates of virus-free and BVF-488-infected *B. cinerea* strains in PDA cultures and the virulence in plant leaves were measured and compared. The phenotype of B05.10 and Pi258.9 strains was different on PDA cultures, however, virus-free and BVF-488 infected B05.10 and Pi258.9 strains showed the same phenotype (Figure 5A,B) indicating the null effect of BVF-V448 replication on the fungal colony morphology of both strains. Additionally, growth rate was faster in the case of B05.10, approximately 1 cm/day, compare with Pi258.9 growth rate of less than 0.5 cm/day. However, no statistically significant differences were observed between virus-free and BVF-V448 infected strains, suggesting that viral infection also has no effect on the growth rate of the fungus, regardless of the infected *B. cinerea* strain (Figure 5A,B).

A pathogenicity test on detached tomato and pepper leaves revealed that both, virus free and BVF-488 transfected *B. cinerea* strain B05.10 induced comparable necrotic lesion diameters at 72 h post inoculation in tomato (around 2.50 cm^2^) and in pepper leaves (around 3.50 cm^2^) (Figure 6A). Smaller lesion sizes were also obtained at 72 h after inoculating both virus free and BVF-488- transfected *B. cinerea* isolate Pi258.9 in tomato leaves (around 1 cm^2^). However, lesions produced by virus-free Pi258.9 were significantly larger (more than 1.5 cm^2^) than those produced by Pi258.9 transfected with BVF-488 (less than 1 cm^2^) on pepper leaves (Figure 6B). These results indicate that the viral infection does not affect the virulence of B05.10 strain on two important hosts as tomato and pepper, and neither the virulence of Pi258.9 on tomato. In contrast, the replication of BVF-V448 significantly decreased the virulence of the Pi258.9 on pepper, its original natural host [28]. The previous infection of Pi258.9 with BcOLV2 isolate BCS12_DN83 and SsULV3, apparently, does not facilitate the infection and transmission of BVF-V448; however, the interaction between the three co-infecting viruses could have a synergistic effect decreasing the virulence of the fungus.

## 4. Discussion

Mycoviral infectious clones are important tools to understand the molecular basis of viral gene function and the study of fungus-virus interaction. In this study, we report the first construction of a stable full-length clone of BVF, a flexuous rod-shaped ssRNA mycovirus, obtained from the plant necrotrophic fungus *B. cinerea* collected from grapevine fields in Spain [29]. Firstly, the complete genomic sequence of the Spanish BVF-V448 was assessed by the determination of the 5′ UTR and 3′ UTR nucleotide sequences and sequencing of the full-length nucleotide sequence. Alignment analysis with the other complete sequences of BVF published revealed an identity between 92% and 84% at the nucleotide level, with the highest identity (92%) with the previous described Spanish sequence LN827953. The differences found between the nucleotide sequences of BVF V448 and LN827953 are likely due to the fact that LN827953 was obtained by NGS directly from the field isolate V448, and BVF V448 was obtained from an independent cloned sequence. However, at the amino acid level, BVF-V448 RdRP and CP shared 97 and 99% identity with LN827953, respectively, indicating that the protein sequences were highly conserved. The 5′ UTR sequences were similar in length, ranging from 63 to 64 nucleotides, with identity values of 87−92% compared to the BVF strains AF238884 and MH338171. The 71 nucleotide 3′ UTR sequences were also highly conserved between BVF isolates with identities ranging from 97% to 100%. Both 5′- and 3′-terminal sequences were predicted with potential to form stem-loop structures that may be important for the regulation of translation of mRNA transcripts and critical for stabilizing RNA viral genomes and initiating genome replication as described previously for other (+)ssRNA virus [56,57,58,59,60].

Howitt and coworkers [34] suggested that BVF CP could be translated via a sgRNA. In this work, we showed that indeed BVF produces one 3′-terminal sgRNA likely involved in the expression of the CP gene. Detection of sgRNAs and identification of their transcription start sites are important for better understanding of the gene expression mechanisms and replication strategies of RNA viruses. Synthesis of 3′-coterminal sgRNAs is the strategy most commonly used by most (+)ssRNA viruses to express their internal genes [61]. Many subgenomic promoters have been mapped in a variety of plant viruses [62,63,64,65,66,67,68,69,70,71,72], but also in fungal viruses [73,74], ranging from over 24 nt [71] to over 100 nt in size [61,62,70]. sgRNA promoters are usually located immediately upstream of the 5′ end of the resulting sgRNA, as first observed in brome mosaic virus (BMV) [75,76]. However, essential components of some sgRNA promoters can be located downstream of the 5′ end of the sgRNA [61,62], very distantly upstream [77], or even on a separate RNA molecule [78]. In this study, the transcription initiation site of the BVF CP sgRNA was mapped at the nt 5822 located in the intergenic region 67 nts downstream the end of the RdRP ORF and 26 nts upstream to the CP ORF. Then, the size of the BVF V448 CP sgRNA promoter (26 nt) is in the range of previous characterized Fusarium graminearum virus-DK21 sgRNA promoters (between 28–30 nt) [73], but shorter than the Mushroom Bacilliform Virus CP sgRNA of 47 nt [74].

BVF CP sgRNA was found to start with a guanine (G), similarly to many other flexiviruses, in which nucleotide substitutions at this position have been shown to significantly reduce virus accumulation in infected plants [68,79]. Indeed, alignment analysis revealed that the hexanucleotide sequence UU**G**AAA, mapped between −2 and +3 nt relative to the transcription site of the BVF CP-sgRNA is 100% conserved between all BVF sequences, and also highly conserved in the corresponding regions among viruses in the families *Alphaflexiviridae*, *Betaflexiviridae*, *Gammaflexiviridae* and *Tymoviridae*. These results strongly suggest that BVF likely share the same gene expression strategy as the viral families mentioned above, employing sgRNA transcription for the translation of its CP, which agrees with the similarities in their genome organizations and viral morphology. This conserved hexanucleotide sequence (UUGAAA) at the transcription start sites of 3′-sgRNAs was shown to be located in the loop of predicted SL structure and might play a regulatory role in recognition by the viral replicase complex on minus strand genomic RNA for the synthesis of plus strand sgRNAs.

The presence of similar or identical sequences at the 5′ ends of genomic RNA (gRNA) and sgRNAs has been reported for several viruses [79,80]. For example, the gRNA and sgRNA of most members of the genus *Potexvirus* within the *Flexiviridae* family starts with a G, and it is likely that this G nucleotide serves as a transcription initiation site of genomic and subgenomic RNAs. In the case of BVF, the transcription initiation site of the CP-sgRNA starts by GA, resembling the 5′-terminal sequence of the BVF genomic RNA, from which infectious transcripts used in this study were produced. Thus, this GA sequence may be important for the synthesis of both, genomic and subgenomic RNA in BVF, as described in other flexivirus [81], and suggest that the minus strand of BVF could be used as template for internal initiation of sgRNA synthesis, yielding a messenger RNA with 3′-ends colinear to the gRNA sequence.

The development of full-length infectious cDNA clones from viral genomes has become the most powerful genetic tool to study the molecular biology of animal, plant, and fungal viruses and to gain important insights into viral infection mechanisms, including determination of viral protein functions and host-virus interactions. In addition, cDNA clone systems have been successfully used for the development of viral-based vectors for gene silencing and expression of foreign proteins, mainly in plant and fungal hosts. Whereas reverse genetic systems have been developed for a large number of animal and plant viruses, only a limited infectious full-length clones have been reported for mycovirus [17,18,19,20,21,22,23,24]. Here, we report the construction and molecular characterization of the first full-length cDNA clone of BVF, from which highly infectious transcripts could be produced as demonstrated by the detection of BVF nucleic acids in transfected *B. cinerea*. We observed that *B. cinerea* strain Pi258.9 accumulate more efficiently the BVF infectious clone, as demonstrated by an earlier detection of BVF nucleic acids by RT-PCR and a higher accumulation of BVF RNA in transfected mycelium compared to B05.10 *B. cinerea* strain. This may be due to the different genetic nature of both strains. Strain B05.10 has been used as a laboratory model for many years for the study of *B. cinerea* and has been used as a recipient for multiple genetic modifications [82]. Moreover, B05.10 strain does not harbor any mycovirus, according to our results using the RNA seq data available in the GenBank, whereas Pi258.9 is a wild strain that already harbors other mycovirus such as Botrytis cinerea ourmia-like virus 2 isolate BCS12_DN83 (MN605468) and Sclerotinia sclerotiorum umbra-like virus 3 isolate SsULV3 (MF444274).

BVF V448 replication had no influence on the phenotype of the fungal strains used, B05.10 and Pi258.9, and the growth rate was similar between the infected and the non-infected strains. Furthermore, the mycovirus cause no effect on the virulence of B05.10, neither in pepper nor in tomato, which are important hosts of *B. cinerea*. Replication of BVF-V448 in Pi258.9 had no effect on its virulence in tomato; however, the virulence of Pi258.9 was decreased by infection with BVF-V448 in its natural host pepper, from which this isolate was originally obtained [28]. It is possible that the hypovirulence is the result of the synergistic interaction between BVF-V448 and the other two mycoviruses infecting Pi258.9, further studies are needed to determine whether different combinations of co-infection by these three viruses contribute to greater hypovirulence. These results showed the absence of influence or decrease in virulence in mixed infections associated with the replication of the infectious clone BVF-V448, which is determinant for the use of BVF infectious clone in studies of plant-*B. cinerea* interaction.

## 5. Conclusions

This report showed the development of infectious synthetic transcripts derived from a full-length cDNA clone of BVF; demonstration of asymptomatic or decreased virulence of BVF infection of two genetically distinct strains of *B. cinerea* using the synthetic transcripts; and detection of BVF sgRNA of corresponding to CP-encoding ORF2. This report provides insight into the gene expression mechanisms of BVF. Understanding the mechanisms of synthesis of mycovirus sgRNA is important for the design of viral vectors capable of inducing rapid silencing of host gene expression in pathogenic fungi. The successful construction of the infectious BVF clone reported in this work will provide an important tool for the future development of a BVF foreign gene expression vector and a BVF-inducing gene silencing vector in *B. cinerea*.

## Figures and Tables

**Figure 1 jof-08-00459-f001:**
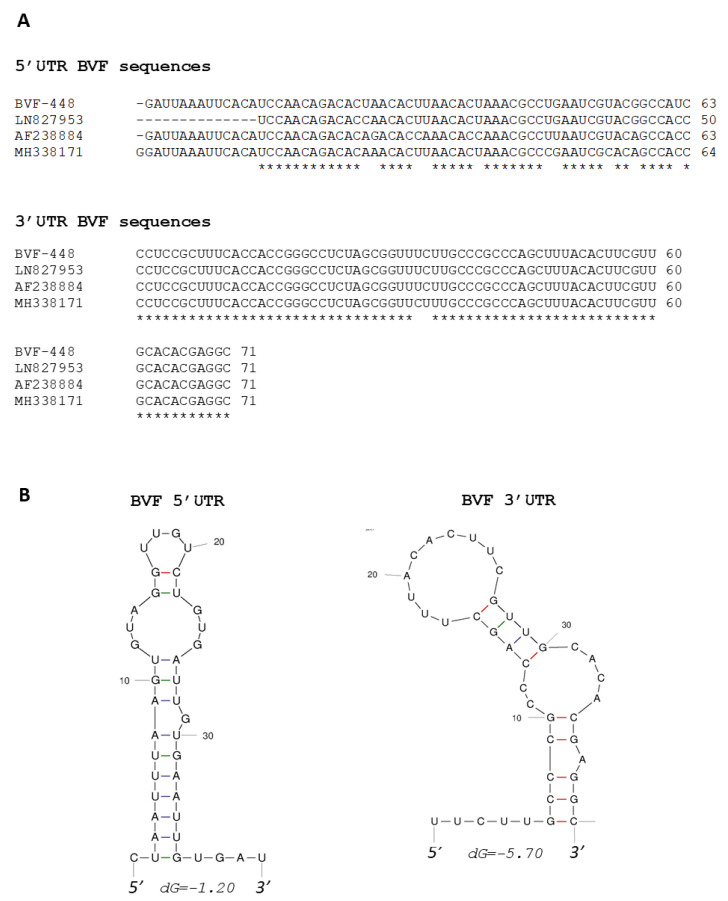
Analysis of 5′ and 3′ UTR sequences of BVF. (**A**) Comparison of sequences in the 5′ UTR and 3′ UTR of different BVF strains; (**B**) Predicted secondary structures of the first 40 nucleotides of the negative strand of 5′ UTR and the last 40 nucleotides of the positive strand of 3′ UTR.

**Figure 2 jof-08-00459-f002:**
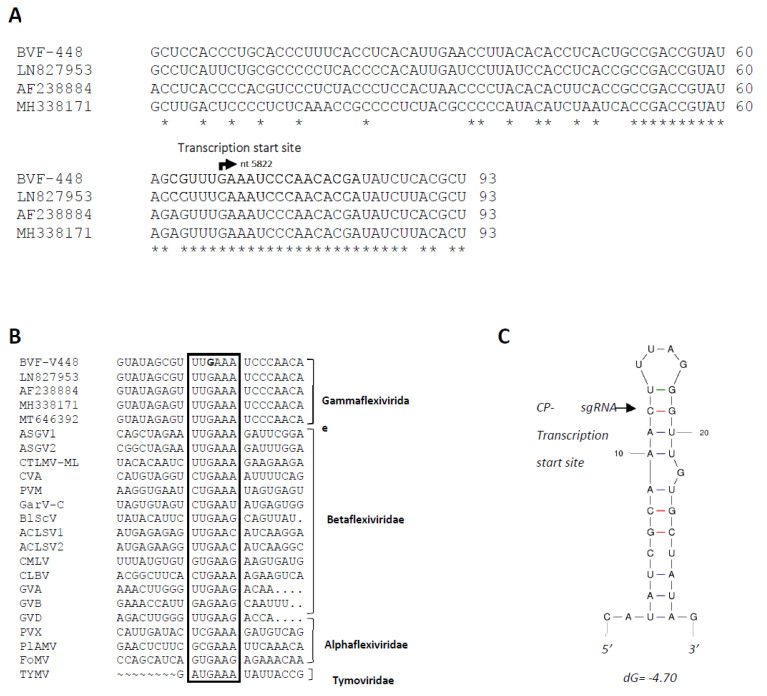
Analysis of BVF CP-sgRNA transcription initiation site. (**A**) Alignment of nucleotide sequences of the genomic RNA from different isolates showing the transcription initiation site of CP-sgRNA of BVF. The bent arrow indicates putative start transcription site at genome nt 5822. Comparison of the Spanish BVF CP sgRNA sequence (OM928008) with those previously published (LN827953, AF238884 and MH338171) and putative promoter sequences for the sgRNAs among other viruses; (**B**) Alignment of the sequence around the transcription start site of different viruses of the families *Tymoviridae*, *Alphaflexiviridae*, *Betaflexiviridae*, and *Gammaflexiviridae* inside the order *Tymovirales*. The highly conserved hexanucleotide is inside a box, with the starting nucleotide of BVF-V448 in bold. (**C**) Predicted secondary structure of the sequence around the BVF CP-sgRNA of BVF. The transcription initiation site of the CP is indicated with an arrow.

**Figure 3 jof-08-00459-f003:**
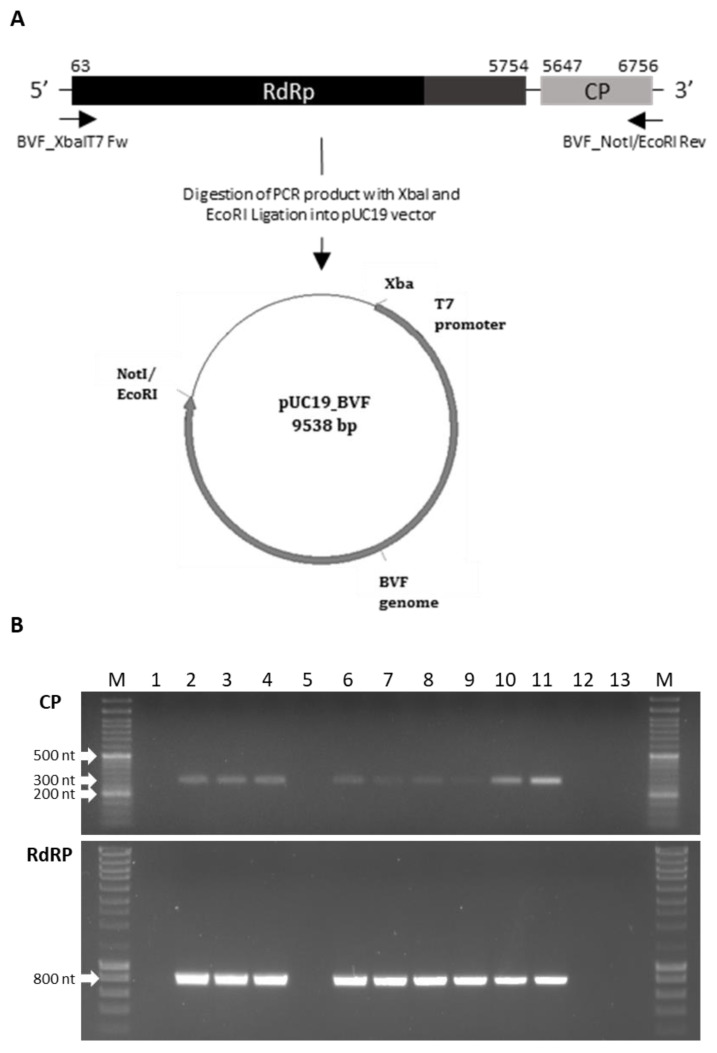
Schematic representation of the construction of the pUC19-BVF clone and detection of BVF in transfected and monosporic strains. (**A**) The entire viral genome (6.8 kb) was amplified using the primers BVF-XbaIT7 Fw and BVF_NotI/EcoRI Rev and ligated into pUC19 between XbaI and EcoRI restriction sites; (**B**) Detection of BVF by RT-PCR using a pair of primers of the CP and RdRp regions in 1. Pi258.9 non transfected strain; 2. Pi258.9A BVF transfected strain; 3. Pi258.9B BVF transfected strain; 4. Pi258.9 monosporic 2 strain obtained from Pi258.9 BVF transfected strain; 5. B05.10 non transfected strain; 6. B05.10 BVF transfected strain; 7. B05.10 monosporic 1 strain obtained from B05.10 BVF transfected strain; 8. B05.10 monosporic 1 strain obtained from B05.10 BVF transfected strain at different state of growth; 9. B05.10 monosporic 2 strain obtained from B05.10 BVF transfected strain; 10. V448 strain natural infected with BVF; 11. BcV36 strain natural infected with BVF; 12 and 13. Negative controls.

**Figure 4 jof-08-00459-f004:**
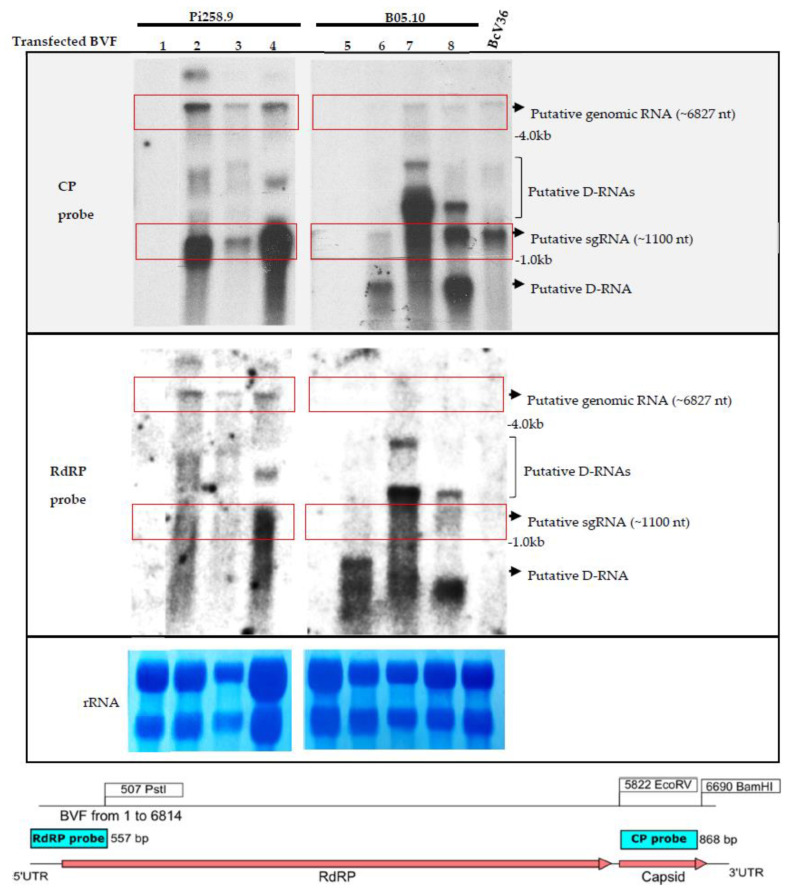
Detection of BVF full length and BVF CP-sgRNA. Northern blot hybridization analysis of total RNA preparations from mock mycelia and *B. cinerea* strain B05.10 and Pi258.9 infected mycelia using a CP and RdRP probes. Total RNA extracts from samples: 1. Pi258.9 non transfected strain; 2. Pi258.9 BVF transfected strain; 3. Pi258.9 monosporic 2 strain obtained from Pi258.9 BVF transfected strain after 5 days of incubation in PDB; 4. Pi258.9 monosporic 2 strain obtained from Pi258.9 BVF transfected strain after 3 days of incubation in PDB 5. B05.10 non transfected strain; 6. B05.10 BVF transfected strain; 7. B05.10 monosporic 1 strain obtained from B05.10 BVF transfected strain after 5 days of incubation in PDB; 8. B05.10 monosporic 1 strain obtained from B05.10 BVF transfected strain after 3 days of incubation in PDB; BcV36 strain natural infected with BVF. Red boxes indicate positions of the bands for BVF genomic RNA (~6827 nt) and putative sgRNA (~1100 nt). Ethidium bromide staining of the gel prior to transfer is shown as loading control (rRNA).

**Figure 5 jof-08-00459-f005:**
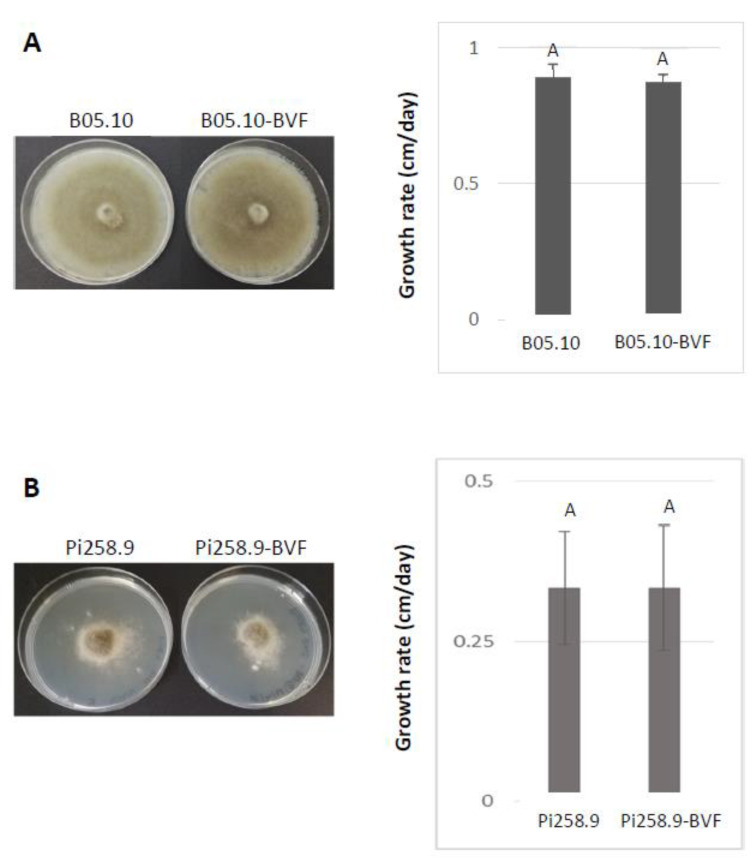
Comparison of phenotype and growth rate of virus-free and BVF-488 –transfected *B. cinerea* isolates B05.10 and Pi258.9. Comparison of colony morphologies and growth rate on PDA (72 h after inoculation) of virus-free and BVF-488—transfected *B. cinerea* strain (**A**) B05.10 and (**B**) Pi258.9.

**Figure 6 jof-08-00459-f006:**
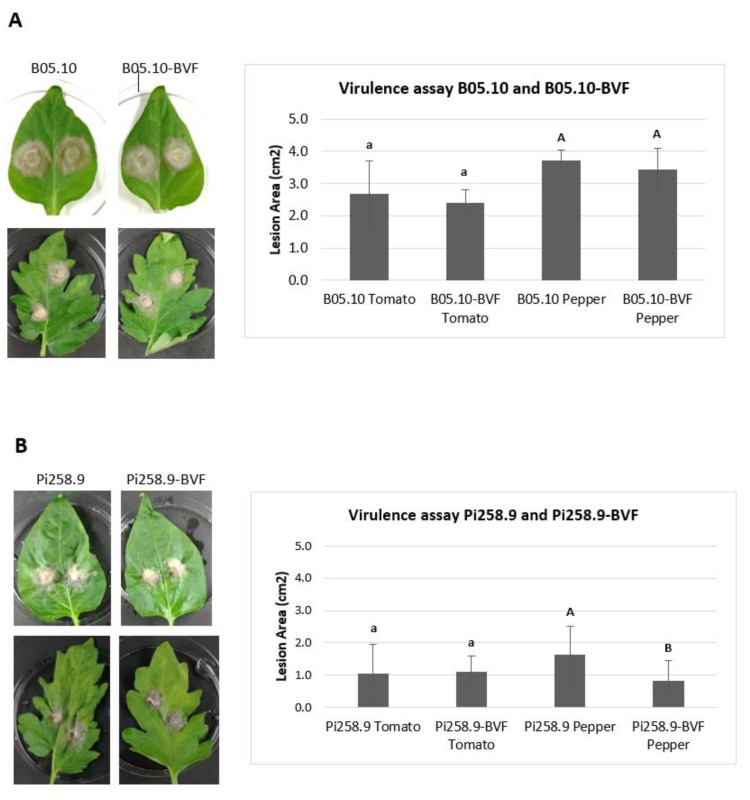
Virulence of virus-free and BVF-488–transfected *B. cinerea* isolates B05.10 and Pi258.9. Comparison of mean lesion area (cm^2^) at 72 h after inoculation with mycelial agar plugs on detached tomato (four replicates) leaves and pepper (seven replicates) leaves of (**A**) B05.10 and (**B**) Pi258.9 strains.

**Table 1 jof-08-00459-t001:** List of primers used in the different steps in the work.

Primer Name	Sequence (5′ to 3′) ^a^	Purpose
BVF_FwQ1	GGCATGGTTGGAACAACCAG	RT-PCR detection
BVF_RvQ2	TCATTCAAGTCGATGCAC	
BVF_XbaIT7 Fw	CTAGCTTCTCTAGATAATACGACTCACTATAGGGGATTAAATTCACATCCAACA	
New BVF738_Rv	CAGGCTCTTCATGGTCATCCAT	
BVF 5out	CTTTGCGGACAGCATATGGTGC	BVF 5’-RACE
BVF 5in	CAGGTATTCGTTTGTGGTAGGGC	
BVF 3out	CCAAAGTGGTCATACGATCACC	BVF 3′-RACE
BVF 3in	CATGGGATACTTTCAGGCATGC	
BVFsg_5out	CCTTACTGGTTGTTCCAACCATGC	Determination transcription initation site CP sgRNA
BVFsg_5in	CGAAAGCTCTTTTCAACGTGGGC	
BVF738_Fw	ATGGATGACTATGAAGAGCCTG	Determination of BVF complete genomic sequence
New BVF738_Rv	CAGGCTCTTCATGGTCATCCAT	
BVF 1550_Fw	CCCGATATGCACTTAAAGCTG	
NewBVF 1550_Rv	CAGCTTTAAGTGCGTATCGGG	
NewBVF 2348_Fw	TCATCCATGACACTGGCAAG	
NewBVF 2348_Rv	CTTGCCAGTGTCATGGATGA	
BVF 3149_Fw	CCAGCGCAGAAGCTGAACAAC	
BVF3628_fw	TATGCCCATGAGTACCGCCGCGTTAGTGAC	
BVF 3628_Rv	GCTGGCGATGGCTTCAGTGG	
New_BVF 3932_Fw	GAACCCTCCGCCTTGTCTAT	
New_BVF 3932_Rv	ATAGACAAGGCGGAGGGTTC	
New_BVF 4755_Fw	GCATCTCACAAGTAAACCGC	
New_BVF 4755_Rv	GCGGTTTACTTGTGAGATGC	
BVF 5279_Rv	GGTCCAAACCTCGCCAGAGA	
New_BVF 6374_Fw	TTGAATCAGCGCACAAAGTCC	
New_BVF 6374_Rv	GGACTTTGTGCGCTGATTCAA	
New_qPCR-intergenic-BVF_For	TTCACCTCACATTGAACCTT	
M13_Rev	gcggataacaatttcacacagg	
M13_For	gtaaaacgacggccagt	
BVF_XbaIT7 Fw	CTAGCTTCTCTAGATAATACGACTCACTATAGGGGATTAAATTCACATCCAACA	Construction of infectious clone
BVF_NotI/EcoRI Rev	TTGAACGGGAATTCGCGGCCGCTTTTTTTTTTTTTTTTTTGCCTCGTGTGCAACGAAG	
BVF_5847 Rv	AGCGTAAGATATCGTGTTGGGAT	
BVF_6756 Fw	CCTCCGCTTTCACCACCG	
BcOLV2-Fw	TGTGCTCAGGAACCGAAGAT	Detection of Pi258.9 mycoviruses
BcOLV2-Rv	GCTTATGAGTTTGGAGGCCG	
SclerotiniaOuMV4-Fw	CTGGCTGACTTTGGTATCGC	
SclerotiniaOuMV4-Rv	CCTCCATCTCTTCTGCCACA	
Umbra SsULV3b Fw	ACGAGCAGTTGGTTCTCAGA	
Umbra SsULV3b Rv	CAACCACAGCCAACATGACA	

## Data Availability

Not applicable.

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
