# Peer review of "Construction and Characterization of a Botrytis Virus F Infectious Clone"

_jof, 2022, doi:10.3390/jof8050459_

Round 1

Reviewer 1 Report

The MS entitled "Effect of infection of the asymptomatic mycovirus Botrytis virus F cDNA clone on the virulence of the pathogenic fungus Botrytis cinerea" by Córdoba et al., construct the full length cDNA clone of BVF. The synthesized RNA transcripts of BVF were introduced into two B. cinerea strains, and the related biological properties were also tested. The MS probably provide a new tool to investigate the interactions between mycoviruses and B. cinerea. However, some issues in the MS are required to be addressed. Generally, the MS is required to be extensively revised and polished.

Major concerns:

  1. This reviewer strongly suggests the authors provide the results of RT-PCR detection of BVF in transfected B. cinerea strains. The authors should try to detect different parts of BVF, not just one region of the genome.
  2. The northern blot should include a positive control like strain V448 for comparison. The label of figure 4 is not clear, as the arrow head position is slightly larger than that of the band. Should the reader think that no signal was detected in the position of BVF genomic RNA? Pi258 and B05.10 were not in the same membrane, it is better for the author to mark it separately. In addition, the author should also provide the blot results with the RdRp probe for comparison, which has been mentioned in the MS.
  3. The quality of the figures is required to be improved, some words in the figures are incomplete. For viral sequence, it is better to use “U” instead of “T” for sequence alignment, the author also mention it is the viral genomic RNA. Figure 5 is rough and not shipshape, and is required for further polished. Why use pepper for B05.10, but pimiento for Pi258.9? Any mean for the green and red color of the bars?
  4. This reviewer doesn’t see the uncropped, untouched, full original images of blots or gels, which is a requirement for the journal.

Minors:

  1. L-12 delete “The mycovirus”.
  2. L-13 “Gammaflexiviridae” should be italic.
  3. L-20, 64 probably other places should be “successful” and “successfully”.
  4. L-78 “spp” should not be italic.
  5. L-80 according to recent literature, it should be far more than 200.
  6. L-118 should be “a key”.
  7. L-176 doesn’t find figure 1S.
  8. L-256 “annuum” should be italic.
  9. L-387 should be mycelia.
  10. L-423 data should be provided.
  11. L-472, 473 probably many other places should be “2.5 and 3.5” not “2,5 and 3,5”, please revise this issue in other places throughout the MS.
  12. L-563-565 any data to support this conclusion?
  13. L-575 should be “mycovirus”.
  14. Any dsRNA was detected?
  15. As no significant effect was observed, the title could be changed to be more concise.

Reviewer 2 Report

Botrytis virus F (BVF) has long been known as a (+)ssRNA virus forming filamentous particles, which infects the very important phytopathogenic ascomycete, Botrytis cinerea. BVF belongs to the species Botrytis virus F (genus Mycoflexivirus, family Gammaflexiviridae). The near-complete or complete genomic sequences have been reported for several BVF isolates. However, most of BVF studies have never gone beyond sequence analyses. The Manuscript ID: jof-1655547 by Cordoba et al. submitted to Fungi describes the establishment of an inoculation method for BVF using synthetic transcripts from full-length cDNA to an isolate BVF-V448. The infectious transcripts were introduced into protoplasts obtained from two BVF-free isolates of B. cinerea. Furthermore, the authors showed the presence of the capsid protein-encoding subgenomic RNA. This finding is of scientific impact given the fact that subgenomic RNA production is known only for a few viruses such as Fusarium graminearum virus 1. Lastly the authors showed BVF to induce no overt phenotypic change in the fungal host, B. cinerea by comparing two pairs of isogenic BVF-free and BVF-infected isolates. 

The manuscript is written and reads well in most parts, with interesting new data. The major points appealing to this reviewer include: 1) development of infectious synthetic transcripts derived from a full-length cDNA clone of BVF, 2) demonstration of asymptomatic infection of two genetically distinct strains of B. cinerea using the synthetic transcripts, 3) detection of BVF subgenomic RNA of corresponding to CP-encoding ORF2. These will definitely advance our understanding of the biology of fungal gammaflexviruses. This reviewer recommends this paper to be accepted for publication in Fungi after appropriate minor revisions, and would like to congratulate the authors on their interesting piece of work. As the authors mention in the paper, this study will open up a new avenue for virus-induced gene silencing and foreign gene expression in the important plant pathogen, B. cinerea. There follow some comments for further improvement:

Minor points:

I feel that the Introduction, Methods and Materials, and Discussion are lengthy.

The text needs to be improved:

Title and other places. The authors cannot technically use the term infectious (cDNA) clone based on the data presented. Infectious cDNA clones refer to those infectious in the form of cDNA such as the ones for CHV1 (Choi and Nuss, 1992) and ScNV23S (Esteban & Fujimura, 2003).

Line 12. Should be “positive-sense, single-stranded.”

Line 39. Hyphenate “negative” and “sense.”

Lines 52. Add “a” before “few.”

Line 327 and 328. Add “the” before “sequence.”

Line 340. Change “to the CP gene” to “of CP ORF,” if the authors mean the 26 upstream of the start codon of the CP gene.

Line 388 - 391. This reviewer thinks that single conidial isolates were to assess vertical transmission but not to assess transfection efficiency.

Line 395. Should be recipient.

Lines 405. Unclear to this reviewer.

Line 563. What data show “more efficient accumulation of BVF?” Do the authors mean that BVF accumulates more in Pi258.9 than in B05.10? It is unclear to this reviewer from Fig. 4 which fungal strain accumulates the virus more, because the two strains appear to have been treated separately in two blots.

I feel that the labellings of some figures lack edges in the pdf downloaded in my computer. For example, the bottom part of 5’ UTR and 3’ UTR is missing in Fig. 1A and 1B. Similar problems can be found in Figs. 2 and 4.

Fig. 4. As it stands, which bands refer to the genomic RNA, subgenomic RNA, and possible defective RNAs is unclear. Label them. It would be better that the authors include gel showing dsRNA patterns and/or genomi RNA encased in particles for the strains transfected and untransfected with the infectious in-vitro synthesized transcripts. The authors need to explain what three lanes with + and one lane with – are. This reviewer wondered why the three lanes shown different northern blot patterns even if they are assumed to be derived from three isolates infected by the same virus BFV.
